# Identification of Novel Genes Involved in Hyperglycemia in Mice

**DOI:** 10.3390/ijms23063205

**Published:** 2022-03-16

**Authors:** Wenke Jonas, Oliver Kluth, Anett Helms, Sarah Voß, Markus Jähnert, Pascal Gottmann, Thilo Speckmann, Birgit Knebel, Alexandra Chadt, Hadi Al-Hasani, Annette Schürmann, Heike Vogel

**Affiliations:** 1Department of Experimental Diabetology, German Institute of Human Nutrition Potsdam-Rehbruecke (DIfE), 14558 Nuthetal, Germany; wenke.jonas@dife.de (W.J.); oliver_kluth@hotmail.com (O.K.); anett.helms@dife.de (A.H.); sarah.voss@dife.de (S.V.); markus.jaehnert@dife.de (M.J.); pascal.gottmann@dife.de (P.G.); thilo.speckmann@dife.de (T.S.); schuermann@dife.de (A.S.); 2German Center for Diabetes Research (DZD), München-Neuherberg, 85764 München, Germany; birgit.knebel@ddz.de (B.K.); alexandra.chadt@ddz.de (A.C.); hadi.al-hasani@ddz.de (H.A.-H.); 3German Diabetes Center (DDZ), Medical Faculty, Institute for Clinical Biochemistry and Pathobiochemistry, Heinrich Heine University, 40225 Duesseldorf, Germany; 4Institute of Nutritional Sciences, University of Potsdam, 14558 Nuthetal, Germany; 5Research Group Genetics of Obesity, German Institute of Human Nutrition Potsdam-Rehbruecke (DIfE), 14558 Nuthetal, Germany; 6Research Group Molecular and Clinical Life Science of Metabolic Diseases, Faculty of Health Sciences Brandenburg, University of Potsdam, 14469 Potsdam, Germany

**Keywords:** β-cell, diabetes, proliferation, apoptosis, QTL

## Abstract

Current attempts to prevent and manage type 2 diabetes have been moderately effective, and a better understanding of the molecular roots of this complex disease is important to develop more successful and precise treatment options. Recently, we initiated the collective diabetes cross, where four mouse inbred strains differing in their diabetes susceptibility were crossed with the obese and diabetes-prone NZO strain and identified the quantitative trait loci (QTL) *Nidd13/NZO*, a genomic region on chromosome 13 that correlates with hyperglycemia in NZO allele carriers compared to B6 controls. Subsequent analysis of the critical region, harboring 644 genes, included expression studies in pancreatic islets of congenic *Nidd13/NZO* mice, integration of single-cell data from parental NZO and B6 islets as well as haplotype analysis. Finally, of the five genes (*Acot12*, *S100z*, *Ankrd55*, *Rnf180*, and *Iqgap2*) within the polymorphic haplotype block that are differently expressed in islets of B6 compared to NZO mice, we identified the calcium-binding protein *S100z* gene to affect islet cell proliferation as well as apoptosis when overexpressed in MIN6 cells. In summary, we define *S100z* as the most striking gene to be causal for the diabetes QTL *Nidd13/NZO* by affecting β-cell proliferation and apoptosis. Thus, *S100z* is an entirely novel diabetes gene regulating islet cell function.

## 1. Introduction

The development of type 2 diabetes (T2D) is multifactorial and has a strong heritable component [1]. Given the global rise in T2D prevalence, there has been a great interest in identifying pharmaceutical target genes for the treatment and prevention of T2D. To this end, several strategies have been developed, including genome-wide association studies (GWAS) and linkage analysis in rodent models, to identify genetic variants associated with T2D and to get further insights into the pathomechanisms of β-cell failure.

Mouse models have proven to be an essential experimental tool in T2D research, and large panels of inbred strains are available that differ in their prevalence toward T2D, thereby providing genetic diversity with respect to diabetes risk [2]. Previously, we initiated the collective diabetes cross, where four inbred strains of mice differing in their diabetes susceptibility were crossed with the obese and diabetes-prone NZO strain to initially identify genomic regions linked to obesity and T2D [3]. Overall, more than 190 partially overlapping genomic regions were detected that are associated with diabetes-related traits, so-called quantitative trait loci (QTL). Among these QTL, the genes such as *Gjb4* (gap junction protein beta 4) and *Ifi202b* (IFN-activated gene 202B) were linked to islet function by regulating islet cell proliferation, apoptosis, and glucose-stimulated insulin secretion [4,5].

The purpose of the current study was to analyze the effect and the underlying genetic cause of the diabetes QTL *Nidd13/NZO* on chromosome 13, a key QTL associated with higher blood glucose levels and increased pancreatic insulin. Finally, functional assays were performed to classify the identified gene variants in the context of β-cell biology.

## 2. Results

Within the German Center for Diabetes Research (DZD), we initiated a cross project using four lean inbred mouse strains with varying T2D susceptibility (diabetes-resistant: B6, 129P2, and C3H; diabetes-prone: DBA), which were crossed with the obese and diabetes-prone NZO mouse strain followed by intensive phenotyping of about 600 offspring of each cross [3]. Genome-wide linkage analysis revealed one major QTL with higher blood glucose levels (Figure 1A) and increased pancreatic insulin on chromosome 13 (*Nidd13/NZO* for non-insulin-dependent diabetes on chromosome 13, derived from NZO) introduced by NZO alleles (Figure 1B). The strongest linkage was detected in the NZOxB6 backcross, but it was also significant for the trait blood glucose in the (NZOxDBA)N2 population. At the genomic peak position (rs4222065, 112.5 Mbp), homozygous NZO (*N/N*) allele carriers showed significantly higher blood glucose levels combined with a decreased body weight compared to heterozygous (*B/N*) mice (Figure 1C). A significant decrease in the insulin content in *Nidd13/NZO^N/N^* mice was detected at the age of 16 weeks. Thus, a gradual decrease in total pancreatic insulin appeared to accompany the increase in fasting plasma glucose levels combined with a fall in fasting plasma insulin levels (Figure 1C).

### 2.1. Characterization of the Phenotype Conferred by the QTL Nidd13/NZO

To analyze the phenotype conferred by the *Nidd13/NZO* locus, congenic male mice harboring one (RCS, *Nidd13/NZO.42.4^B/N^*) or two (*Nidd13/NZO.42.4^B/B^*) alleles from B6 on chromosome 13 (78.0–120.4 Mbp) on NZO background were characterized and compared to homozygous NZO (*Nidd13/NZO.42.4^N/N^*) controls on a high-fat diet (HFD, Figure 2). On HFD, body weight and body fat content were lower, reaching significance at week 10, in *Nidd13/NZO.42.4^N/N^* mice compared to *B/B* controls (Figure 2). Even with lower body weight and fat mass, *Nidd13/NZO.42.4^N/N^* mice showed slightly increased blood glucose levels at week 8 and 10 in comparison to *Nidd13/NZO.42.4^B/B^* mice (Figure 2), whereas no differences in pancreatic insulin content were detected between the genotypes (data not shown). In summary, introgression of the diabetes QTL *Nidd13/NZO* on the NZO background has only a mild effect on the increase in the blood glucose levels, however, even under conditions of the lower degree of body weight.

### 2.2. Fine Mapping of the Critical Diabetogenic Interval of Nidd13/NZO

The region spanning 42.4 Mbp (78.0–120.4 Mbp) of the *Nidd13/NZO* locus harbors 644 genes, including 377 gene models, 8 miRNAs, and 51 Riken genes [6]. According to the hypothesis that the causal gene variant(s) are different between NZO and B6, a haplotype map was created of the putative critical region on chromosome 13 based on NZO/HiltJ and C57BL/6J (B6) SNP information from the Wellcome Trust Sanger Institute [7,8]. The number of polymorphic SNPs between NZO and B6 was determined for each 250 kb window [9]. Regions exceeding a threshold of 100 SNPs/window were considered polymorphic according to NZO ≠ B6 (Figure 3A). Only genes fulfilling these criteria were considered as candidates, reducing the number to 111 genes.

Moreover, gene expression profiles of pancreatic islets were generated from parental and congenic mice carrying the 42.4 Mbp fragment of the *Nidd13/NZO* locus. Eight genes within the critical region were differentially expressed (LFC > ±1) in pancreatic islets of B6 compared to NZO mice (Figure 3B), whereas five of these genes (highlighted in red) were also different between homozygous *Nidd13/NZO.42.4^N/N^* mice compared to homozygous *Nidd13/NZO.42.4^B/B^* controls (Figure 3C).

In previous studies, we analyzed the genome-wide expression pattern in islets of the diabetes-prone NZO and the diabetes-resistant B6-*ob/ob* mice to obtain more information about genes involved in the pathogenesis of T2D [10]. We found that B6-*ob/ob* mice are able to compensate for higher blood glucose levels by increasing the β-cell mass, whereas the NZO strain develops hyperglycemia due to β-cells loss [5,10]. As the diabetes locus on chromosome 13 is also associated with higher blood glucose levels and reduced pancreatic insulin content, we screened the expression of the five candidate genes in the NZO vs. B6-*ob/ob* data set. As shown in Figure 3D, all genes except *Acot12* revealed a differential expression between both strains in pancreatic islets. To get further information about the expression of the five candidate genes in the different endocrine islet cell subtypes, we also implemented single-cell data of islets from diabetes-susceptible NZO and the diabetes-resistant B6-*ob/ob* mice. In addition to α, δ, and γ cells, we defined six specific β-cell clusters which differ in their expression profile and their response to glucotoxic conditions [11]. In general, the genes *S100z* and *Iqgap2* showed high expression levels in β-cell clusters (*Beta1-Beta4*). Interestingly, the highest expressions for *S100z* and *Iqgap2* were detected in the proliferating β-cells, the *BetaP* (*P = proliferation*) cluster, which is defined by high expression of the marker gene *Mki69* (marker of proliferation Ki-67). This suggests a possible role of both genes in islet-cell proliferation (Figure 3E).

In total, we identified the genes *S100z* (S100 calcium-binding protein Z), *Iqgap2* (IQ motif containing GTPase activating protein 2), *Rnf180* (ring finger protein 180), and *Ankrd55* (ankyrin repeat domain 55) as potential novel regulators of β-cell function.

### 2.3. Impact of the Different Nidd13/NZO Genes on Islet Cell Proliferation and Apoptosis 

To clarify the impact of the different candidate genes of the *Nidd13/NZO* locus on islet cell function, we overexpressed each gene in mouse insulinoma 6 (MIN6) cells, a β-cell line, and studied their proliferation and apoptosis. The different genes were overexpressed via adenoviral-mediated infection (Figure 4A), and proliferation capacity was examined by analyzing BrdU incorporation. Cells infected with an empty virus were used as control. Interestingly, overexpression of *S100z* decreased the number of BrdU positive cells (33.0% ± 0.5 in comparison to 37.0% ± 1.6 of control cells), suggesting that *S100z* impairs the proliferation capacity of MIN6 cells (Figure 4B,C). Overexpression of the other *Nidd13/NZO* candidate genes in MIN6 cells had no impact on proliferation (Figure 4B,C; Ad-*Ankrd55*: 39.1% ± 2.1; Ad-*Iqgap2*: 38.3% ± 2.6; Ad-*Rnf180*: 36.8% ± 3.3). To verify whether S100Z is indeed affecting the proliferation capacity of pancreatic islets, we overexpressed this gene in primary islet cells derived from NZO mice, which exhibit a low endogenous S100Z expression (Figure 4D). Overexpression of *S100z* leads to a significantly reduced proliferation, which further confirms our previous observations in MIN6 cells.

Previous studies showed that the loss of β-cell mass due to apoptosis plays an important role in the development of T2D [5]. Therefore, apoptosis was induced by lipotoxic conditions (0.4 mM palmitate) in MIN6 cells overexpressing one of the *Nidd13/NZO* candidate genes. Detection of cleaved caspase-3 as a marker of apoptosis by Western blotting revealed that apoptosis was significantly increased by 64% in cells infected with the control virus under these conditions (Figure 5A). In contrast, MIN6 cells overexpressing *S100z* showed no increase after induction of apoptosis with palmitate compared to basal conditions, clearly indicating that the gene *S100z* has an impact on cell apoptosis (Figure 5A). Overexpression of *Ankrd55*, *Rnf180,* and *Iqgap2* in MIN6 cells resulted in a similar increase in cleaved caspase-3 levels compared to cells infected with the control virus (Figure 5B). To further investigate the effect of *S100z* on apoptosis, we also overexpressed this gene in primary islets derived from NZO mice and analyzed the expression of the ER stress-induced genes *Atf4* and *Edem1* as well as the apoptotic genes *Bid* and *Chop*. As shown in Figure 5C, apoptosis was increased under lipotoxic conditions in cells infected with the control virus. The palmitate-induced induction of *Edem1*, *Bid,* and *Chop* was significantly reduced in *S100z*-overexpressing cells compared to controls, whereas *Atf4* expression was not affected.

In summary, *S100z* affects both proliferation and apoptosis and may therefore serve as a causative gene for the *Nidd13/NZO* phenotype.

## 3. Discussion

In the present study, we used the linkage data of the collective diabetes cross to gain new insights into the genetic components regulating β-cell failure. We identified the QTL *Nidd13/NZO* for diabetes-related traits on chromosome 13 and combined bioinformatic analysis with expression data to identify novel diabetes genes. Subsequent functional analysis of the impact of selected candidate genes revealed *S100z* as the most likely candidate to regulate β-cell function.

The diabetogenic allele of the *Nidd13/NZO* on chromosome 13 induces hyperglycemia in the NZOxB6 backcross population combined with a decrease in the insulin content at later time points. In fact, introgression of the diabetes QTL on the NZO background moderately increased blood glucose levels, despite a lower degree of adiposity of the mice carrying the *Nidd13/NZO* allele of NZO mice. However, the effect on blood glucose was not significant. This might be explained by the obese background of the congenic mice (*Nidd13/NZO.42.4*) with more than 99% NZO genome. This background is inducing higher blood glucose values also in recombinant congenic mice carrying the homozygous B6 allele of *Nidd13/NZO* already at an early time point (week 6, Figure 2). Thus, it is likely that the selective and additional effect of the *Nidd13/NZO* locus was masked by the presence of other diabetes genes. 

For the detection of *Nidd13/NZO* candidate genes, we took advantage of the genetic architecture of widely used inbred mouse strains that share common genetic ancestry [3,12]. By establishing a computational framework combining linkage data from different backcross populations and by superimposing strain-specific haplotype information and genome-wide expression data, we considerably reduced the number of candidate genes within the critical genomic interval linked to diabetes-related traits. Of the five genes within the haplotype block that are differently expressed in islets of homozygous B6 compared to homozygous NZO mice, we identified the *S100z* gene to affect islet cell proliferation as well as apoptosis. 

The S100 proteins belong to a calcium-binding cytosolic protein family, which compose of more than 20 known members [13]. S100 family members have a broad range of intracellular and extracellular functions that encompass the regulation of cell apoptosis, proliferation, differentiation, migration, energy metabolism, calcium balance, protein phosphorylation, and inflammation [14,15,16,17]. In addition, S100 proteins may contribute to the development of many types of malignant tumors, autoimmune diseases, and chronic inflammatory diseases [18]. In the present study, we have shown that the family member *S100z*, located within the diabetes QTL *Nidd13/NZO*, is involved in β-cell proliferation as well as apoptosis. A common feature of S100 proteins is their ability to interact with the tumor suppressor p53, and they have been shown to both potentially activate and inhibit p53 [19]. Certain S100 proteins, namely S100A4, S100A8/S100A9 heterodimer, and S100B, have been implicated in the pathophysiology of obesity-promoting macrophage-based inflammation via toll-like receptor 4 and/or receptor for advanced glycation end-products ligation [20]. Additionally, serum levels of S100A4, S100A8/S100A9, S100A12, and S100B correlate with insulin resistance/T2D, metabolic risk score, and fat cell size [20]. It was already shown that S100A8 is an endogenous islet-derived secretory peptide that is induced by a combination of infiltrating macrophages, palmitate (lipotoxicity), and high glucose (glucotoxicity), resulting in the activation of macrophages and potentiation of islet inflammation and β-cell death through a positive feedback loop [21].

S100Z is downregulated in several tumors [22], but no functional roles have been reported so far. However, given the known function of other S100 members, it is likely that S100Z also has an impact on β-cell function. In previous studies, we have shown that a carbohydrate intervention (+CH) in obese NZO mice increases blood glucose levels and induces islet-cell apoptosis within a few days. By contrast, B6-*ob/ob* mice, which carry a leptin mutation on the B6 background, do not develop hyperglycemia under +CH conditions due to induction of β-cell proliferation [5,10]. Therefore, it could be speculated that the identified gene *S100z,* together with other genes, is causal for this difference. 

Although the functional assays suggest that the *S100z* gene could be causal for the diabetogenic effect of the QTL *Nidd13/NZO*, it is also likely that one of the other genes with a differential expression within the critical haplotype block, *Ankrd55*, *Rnf180,* or *Iqgap2*, can also influence β-cell function. For instance, knockout mouse models of the gene *Iqgap2*, an IQ motif containing GTPase activating protein, implicate a specific role of this gene in glucose homeostasis. *Iqgap2*-deficient mice demonstrated metabolic inflexibility, fasting hyperglycemia, and obesity. Such phenotypic characteristics were associated with aberrant hepatic regulations of glycolysis/gluconeogenesis, glycogenolysis, lipid homeostasis, and futile cycling corroborated with corresponding proteomic changes in cytosolic and mitochondrial compartments [23]. Interestingly, *Ankrd55* has also already been shown to be implicated in diabetes development as the human *ANKRD55* locus is associated with both body composition and diabetes [24,25]. The *ANKRD55* locus showed strong associations with XXLVLDL particle response, and Li-Gao et al. speculate that the *ANKRD55* locus could potentially affect chylomicron synthesis and transportation after fat intake and consequently influence β-cell function and ultimately lead to a higher risk of T2D [25].

Moreover, it is also reasonable to assume that a number of genetic variants, including noncoding SNPs, indels, copy number polymorphisms, and yet unknown de novo mutations, may exert effects on regulatory circuits in the locus, thereby affecting islet cell function and glycemic control [26]. Consequently, future studies are needed to determine the contribution of *S100z*, the other *Nidd13/NZO* candidate genes, as well as genetic variants within the locus on β-cell function. A second limitation of our study is the lack of validation of the effects of S100Z in human islets.

In summary, with the current study, we identified the calcium-binding protein *S100z* as the most striking gene to be causal for the diabetes QTL *Nidd13/NZO* by affecting β-cell proliferation as well as apoptosis and is thereby regulating glucose homeostasis. On these lines, it would be interesting to investigate whether alterations in *S100z* gene expression influence other metabolic pathways in the β-cell, e.g., insulin-stimulated glucose uptake, and to further clarify the exact mechanism of how *S100z* regulates these various processes within pancreatic islets.

## 4. Materials and Methods

### 4.1. Animals

The N2 backcross population was generated as previously described [3]. Briefly, female NZO mice from our own breeding colonies (NZO/HIBomDife) [27] were mated with male DBA/2J (Jackson Lab, Bar Harbor, ME, USA), C57BL/6JRI (B6; Janvier Laboratories, Le Genest St. Isle, France), C3H/FeJ (C3H; Helmholtz Center Munich, Germany) or 129P2/OlaHsd (129P2; German Institute of Human Nutrition, Nuthetal, Germany) mice to produce F1 hybrids. Male F1 mice from each cross were subsequently backcrossed to female NZO to produce N2 mice that were metabolically characterized. Mice were housed in type 2 or type 3 macrolon cages with bedding made of soft wood shavings (spruce) in groups of 3–6 and kept under standard conditions (conventional germ status, 22 °C with 12 h light/dark cycling). After weaning at 3 weeks of age, animals were placed on a high-fat diet (HFD, D12451 Research Diets Inc., New Brunswick, NJ, USA), containing 45%, 35%, and 20% kcal from fat, carbohydrate, and protein, respectively. Recombinant congenic mice were bred by repeated backcrossing of male mice selected for the *Nidd13/NZO* locus (*N/N* or *B/B*) with NZO females, and male offspring were characterized in the N9 generation on HFD. Male NZO/HIBomDife and B6.V-Lep*^ob/ob^*/JBomTac (B6-*ob/ob*) mice (Charles River Laboratories, Sulzfeld, Germany) were used and housed as described [5]. All experiments were approved by the ethics committee of the State Agency of Environment, Health and Consumer Protection (Federal State of Brandenburg, Potsdam, Germany).

### 4.2. Body Composition

Body composition was determined by non-invasive nuclear magnetic resonance spectroscopy (EchoMRI™-100 system, Echo Medical Systems, Houston, TX, USA).

### 4.3. Blood Glucose

Blood glucose was measured in the morning between 8 and 10 am with a CONTOUR^®^ XT glucometer (Bayer Consumer Care AG, Leverkusen, Germany).

### 4.4. Plasma Analysis

Insulin concentrations were determined by ELISA (ALPCO, Salem, MA, USA).

### 4.5. Pancreatic Insulin Content

The whole pancreas was homogenized in ice-cold acidic ethanol (0.1 mol/L HCl in 70% ethanol) and incubated overnight at 4 °C. After centrifugation (16,000 × *g*, 10 min), insulin content was measured in the supernatant fraction with the Mouse High Range Insulin ELISA (ALPCO).

### 4.6. Genotyping

Genomic DNA was extracted from mouse tail-tips using the Invisorb Genomic DNA Kit II (STRATEC Molecular GmbH, Berlin, Germany), following the manufacturer’s instructions. Competitive allele-specific PCR (KASP) genotyping of the different NZO backcross mice was performed by LGC genomics (LGC group, Teddington, UK). Recombinant congenic mice containing the *Nidd13/NZO* locus were genotyped by KASP assays (LGC) or by PCR with oligonucleotide primers obtained from Sigma (St. Louis, MO, USA), and the microsatellite length was determined by non-denaturing polyacrylamide gel electrophoresis.

### 4.7. Linkage Analysis

Genome-wide linkage analysis was performed as previously described [3]. Briefly, genetic map, genotyping errors, and linkage between individual traits and genotypes were assessed with the software package R/qtl (version 1.04-8) using the expectation maximization (EM) algorithm and 1000 permutations [28].

### 4.8. RNA Extraction and Genome-Wide Expression Analysis

Primary islet cells of congenic and NZO mice were isolated and cultivated as described [5].

Isolation of total RNA from islets of Langerhans and MIN6 cells was performed with the miRNeasy Micro Kit from (QIAGEN, Hilden, Germany). RNA quality was determined using an Agilent 2100 Bioanalyzer (Agilent Technologies, Santa Clara, CA, USA), and the manufacturer’s instructions were followed to measure RNA integrity (RIN).

Genome-wide expression analyses of parental mice were performed with 150 ng RNA according to the Ambion WT Expression Kit and the WT Terminal Labeling Kit (Thermo Fisher Scientific, Darmstadt, Germany). All protocol steps were monitored using an RNA 6000 nano kit (Agilent). Mouse Gene 1.0 ST arrays were hybridized with labeled complementary RNA samples. Data were collected with the GeneChip scanner 3000 7G, and analyses of primary data were performed with the GDAS 1.4 package, [Affymetrix, (Thermo Fisher Scientific)]. Data were analyzed with Expression ConsoleTM v1.1 and Transcriptome Analysis ConsoleTMv2.0 (Affymetrix) as previously described [29].

Genome-wide expression analysis of B6-*ob/ob* and NZO mice was performed by GATC (Konstanz, Germany) on an Illumina HiSeq platform with five samples (each containing islets of one to two mice) for each group. Quality control and bioinformatic analysis were described previously [30].

### 4.9. Quantitative Real-Time PCR

RNA of MIN6 cells and RNA of isolated islets from congenic (*Nidd13/NZO*; *B/B*, *N/N*) and parental NZO mice was reversed transcribed (M-MVL RT, Promega, Madison, WI, USA) for quantitative real-time PCR. Genes of interest were detected using specific TaqMan (Thermo Fisher Scientific, Waltham, MA, USA) and IDT (Integrated DNA Technologies, Coralville, IA, USA) probes. Expression levels were evaluated using the 2(-Delta C(T)) method [31] with β-actin (*Actb*), eukaryotic translation elongator factor 2 (*Eef2*), or TATA box binding protein (*Tbp*) as internal controls.

### 4.10. Single-Cell Data

Libraries were pooled and sequenced on an Illumina NovaSeq6000 (Illumina, San Diego, CA, USA) with an average read depth of 60,000–90,000 reads/cell. Read filtering and counting (barcode and unique molecular identifiers) were performed with the CellRanger-Pipeline (version 3.0.2) provided by 10X Genomics (Pleasanton, CA, USA). Default parameters of CellRanger were used to detect high-quality barcodes, which are based on the overall distribution of total UMI counts. Further downstream analysis, quality control, and correction for batch effect were performed using Scanpy [11,32]. Plotting was done with the “scanpy.pl.DotPlot” of Scanpy.

### 4.11. Cell Culture

Mycoplasma-free mouse insulinoma 6 (MIN6) cells were cultured in Dulbecco’s modified Eagle’s medium (P04-03590, PAN-Biotech, Aidenbach, Germany) containing 10% heat-inactivated fetal calf serum (Life Technologies, Darmstadt, Germany) in a humidified incubator at 37 °C and 5% CO_2_.

### 4.12. Overexpression of Candidate Genes in MIN6 and Primary Islet Cells and BrdU Proliferation Assay

For overexpression, MIN6 cells were infected with either an adenovirus carrying cDNA from *S100z* (Ad-*S100z*, MOI 100), *Iqgap2* (Ad-*Iqgap2*, MOI 1000), *Rnf180* (Ad-*Rnf180*, MOI 1000), *Ankrd55* (Ad-*Ankrd55*, MOI 1000), or the empty virus (Ad-∅, MOI 100 or MOI 1000); (Ad-mS100z-Myc: 20150512T#5; Ad-CMV-mIqgap2-Myc: 20170509T#1; Ad-mRnf180-Myc: 20160104T#4; Ad-mAnkrd55-Myc: 20150519t#8; Ad-CMV-NULL: 20150623t#7, Vector Biolabs, Malvern, PA, USA). Primary islet cells were dispersed and infected with an adenovirus carrying cDNA from *S100z* (Ad-*S100z*, MOI 100) or the empty virus (Ad-∅, MOI 100). Infected MIN6 and primary islet cells were incubated in the viral medium for 24 h, and then, MIN6 cells were labeled with BrdU (100 μmol/L) for 2 h and primary islet cells for 48 h. Cells were fixed with 4% paraformaldehyde. Cell membranes were permeabilized (0.2% saponin), DNA was denatured (2 M HCL), histones were eliminated (0.1% trypsin), and cells were incubated with primary antibodies against BrdU (1:800, Abcam, Cambridge, UK) overnight at 4 °C. Detection was performed with fluorophore-labeled secondary antibody (rat: Alexa Fluor546, 1:400, Invitrogen, Carlsbad, CA, USA) and DAPI (1:1000, Roche, Basel, Switzerland) for 1 h at RT and documented with the BZ900 Fluorescence Microscope (Keyence Deutschland GmbH, Neu-Isenburg, Germany). Statistics were performed by blinded quantification of 10 photographs of at least two coverslips per infection by using the ImageJ software package [v1.52; Wayne Rasband (NIH)]. Induction of apoptosis and Western blot analysis were conducted. 

MIN6 cells were seeded in a 12-well plate and infected with Ad-*S100z* (MOI 100), Ad-*Iqgap2* (MOI 1000), Ad-*Rnf180* (MOI 1000), Ad-*Ankrd55* (MOI 1000), or the empty virus (Ad-∅, MOI 100 or MOI 1000) for 24 h. Primary islet cells were seeded in a 12-well plate and infected with Ad-*S100z* (MOI 100) or the empty virus (Ad-∅, MOI 100) for 24 h. Subsequently, cells were treated with 5.5 mM glucose and 0.4 mM palmitate for 48 h, washed with PBS and lysed in 70 μL lysis buffer per well (20 mM Tris-HCL, 150 mM NaCl, 1 mM EDTA, 1 mM EGTA, 1% Triton (pH 7.4), 1× protease inhibitor cocktail (Roche) and 1× phosphatase inhibitor cocktail (2.5 mM Na_4_P_2_O_7_, 1 mM C_3_H_7_Na_2_O_6_P, 1 mM Na_3_VO_4_, 1 mM NaF, Roche). Western blotting in MIN6 cells was performed by using a primary antibody against cleaved caspase 3 (1:500, Cell Signaling Technology, Danvers, MA, USA) and as loading control α-tubulin (1:500, Sigma, St. Louis, MO, USA) followed by application of secondary DyLight-680 conjugated goat anti-mouse antibody (1:20,000, Invitrogen, Invitrogen) and DyLight-800 conjugated goat anti-rabbit antibody (1:10,000, Thermo Fisher Scientific).

### 4.13. Statistical Analysis

Statistical analysis was performed by either unpaired Student’s *t*-test or one-way analysis of variance (ANOVA) followed by post hoc Bonferroni test as appropriate. Statistical analyses were conducted using the GraphPad 8.0 (GraphPad Software, San Diego, CA, USA). A *p*-value < 0.05 was considered significant, and values are expressed as means ± SEM.

## Figures and Tables

**Figure 1 ijms-23-03205-f001:**
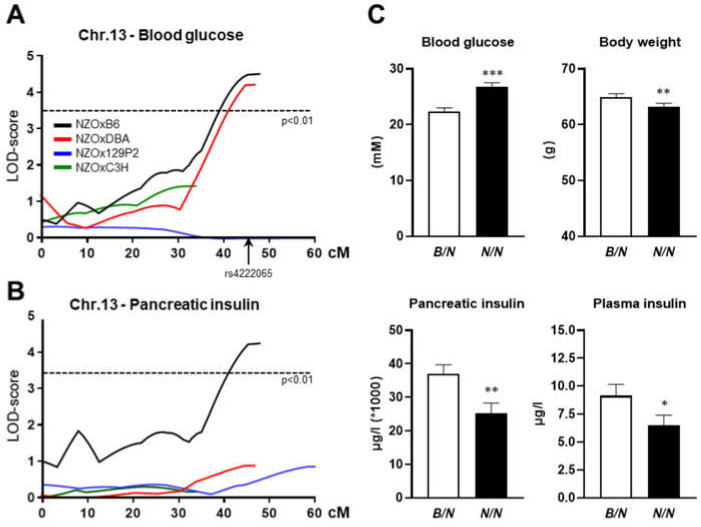
Physical maps of the QTL *Nidd13/NZO* identified by genome-wide linkage analysis of NZO backcross populations. Genome-wide linkage analysis for the traits blood glucose (**A**) and total pancreatic insulin (**B**) revealed a single QTL on chromosome 13. The analysis was performed with data from male mice of the different N2 populations. The horizontal line indicates the threshold of significance (*p* < 0.01). (**C**) Effect sizes of traits associated with the QTL *Nidd13/NZO* in the (NZOxB6)N2 cohort (*N/B*: *n* = 148; *N/N*: *n* = 139). Data are presented as means ± SEM. * *p* < 0.05; ** *p* < 0.01; *** *p* < 0.005.

**Figure 2 ijms-23-03205-f002:**
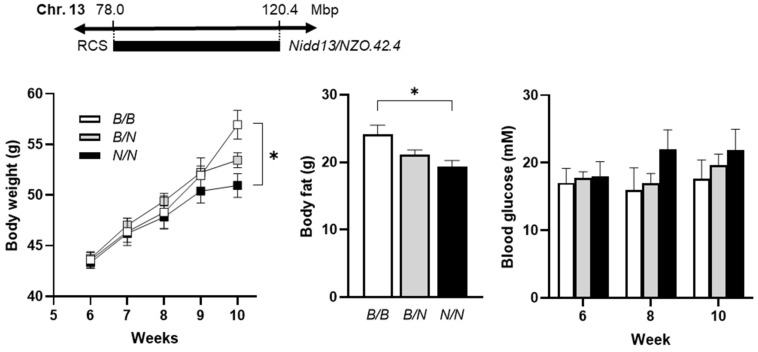
Phenotypic traits of congenic *Nidd13/NZO* mice. Male mice carrying 42.4 Mbp of chromosome 13 from NZO or B6 (*N/N*, *N/B*, *B/B*) on the NZO background were characterized for the development of body weight, body fat, and blood glucose on high-fat diet until 10 weeks of age. Data are presented as means ± SEM. * *p* < 0.05; *B/B*, *n* = 8–11; *N/B*, *n* = 31–33; *N/N*, *n* = 8–11.

**Figure 3 ijms-23-03205-f003:**
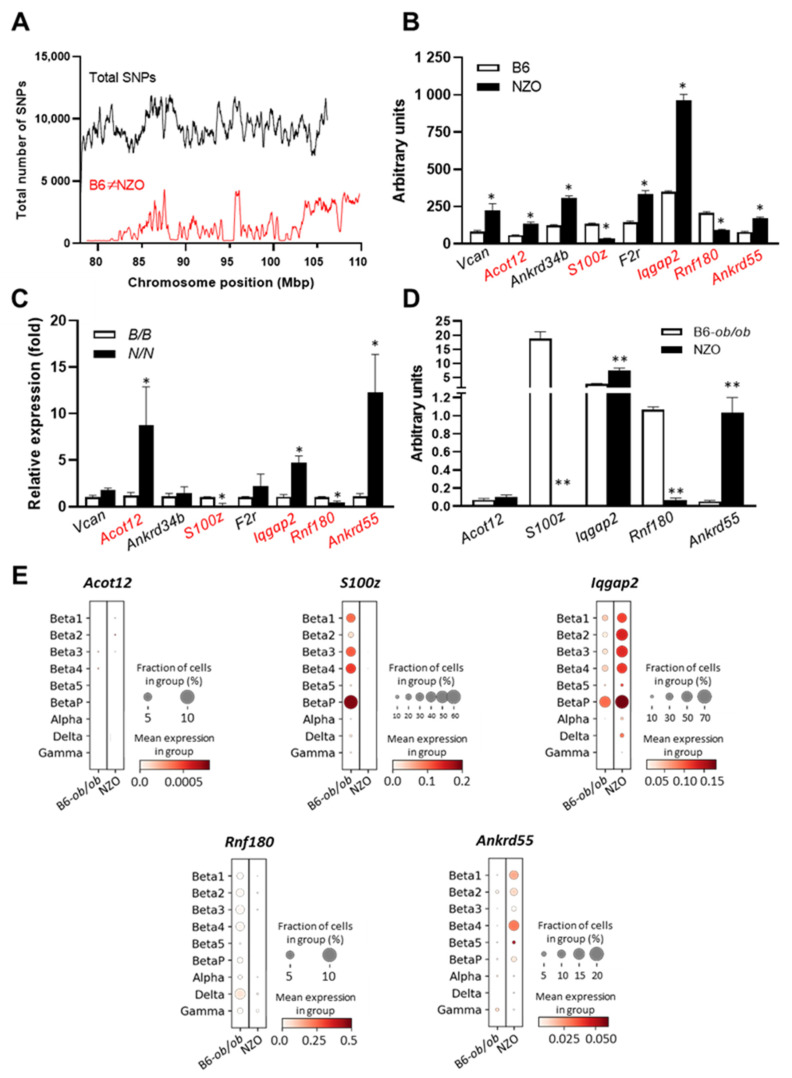
Combined approach of haplotype and gene expression analysis in islets for the identification of gene variants within the *Nidd13/NZO* locus. (**A**) Haplotype map of single nucleotide polymorphisms (SNPs) within the critical region of *Nidd13/NZO* (78.0–120.4 Mbp). Black line: total number of SNPs (all SNPs annotated for the B6 reference genome). Red line: SNPs according to B6≠NZO. (**B**) Genes within the critical *Nidd13/NZO* interval displaying a differential expression in pancreatic islets of parental mice, (**C**) *Nidd13/NZO.42.2* congenics, and (**D**) B6-*ob/ob* versus NZO mice. (**E**) Single-cell data of *Nidd13/NZO* candidate genes. Dot plot representing expression levels by color code and fraction of cells expressing the gene. Expression levels are depicted from white to red (low to high), and dot size presents fraction of cells expressing the indicated gene. * *p* < 0.05; ** *p* < 0.005.

**Figure 4 ijms-23-03205-f004:**
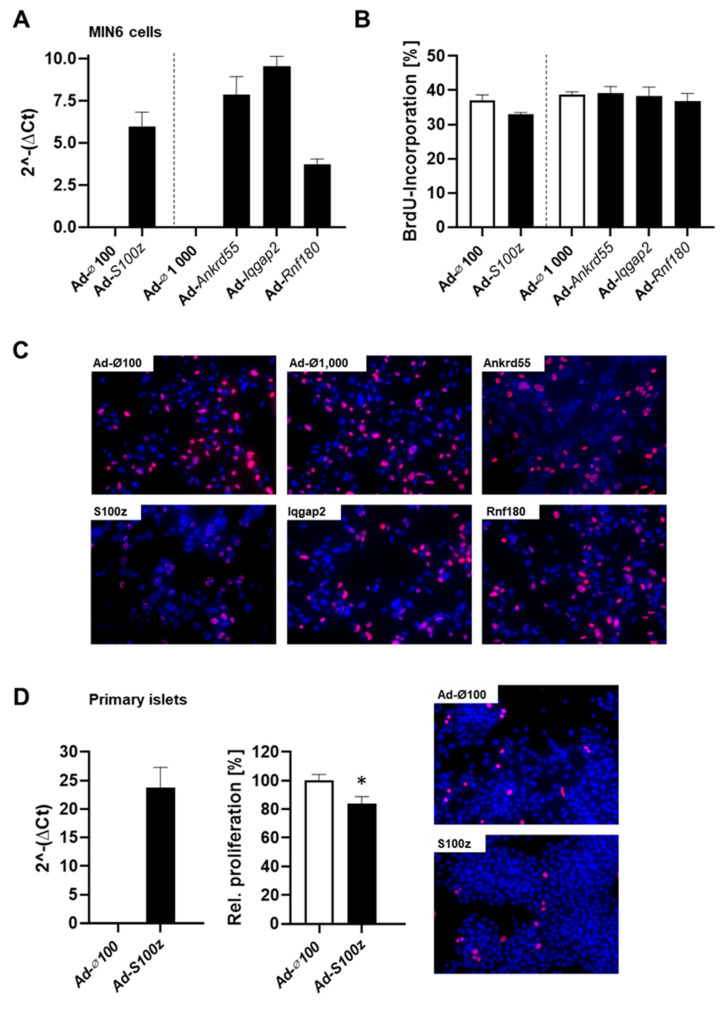
Impact of the different *Nidd13/NZO* candidate genes on proliferation. (**A**) MIN6 were infected with an empty adenovirus (Ad-∅) as control or an adenovirus expressing one of the *Nidd13/NZO* candidate genes; overexpression was analyzed by qRT-PCR. (**B**) Infected MIN6 cells were incubated with BrdU for 72 h. For quantification, cells were counted and assessed as BrdU positive or negative. For each group, 10 pictures of at least two coverslips per infection (*n* = 4) were evaluated. (**C**) Representative immunocytochemical co-stainings of BrdU (magenta) and DAPI (blue). (**D**) Primary islets of NZO mice were dispersed and infected with an empty adenovirus (Ad-∅) as control or an adenovirus expressing *S100z,* and overexpression was analyzed by qRT-PCR. After incubation with BrdU for 48 h, islets were co-stained with a BrdU (magenta) and DAPI (blue) antibody. For each group, at least 10 images of four or two coverslips per infection (*n* = 2) were evaluated. * *p* < 0.05.

**Figure 5 ijms-23-03205-f005:**
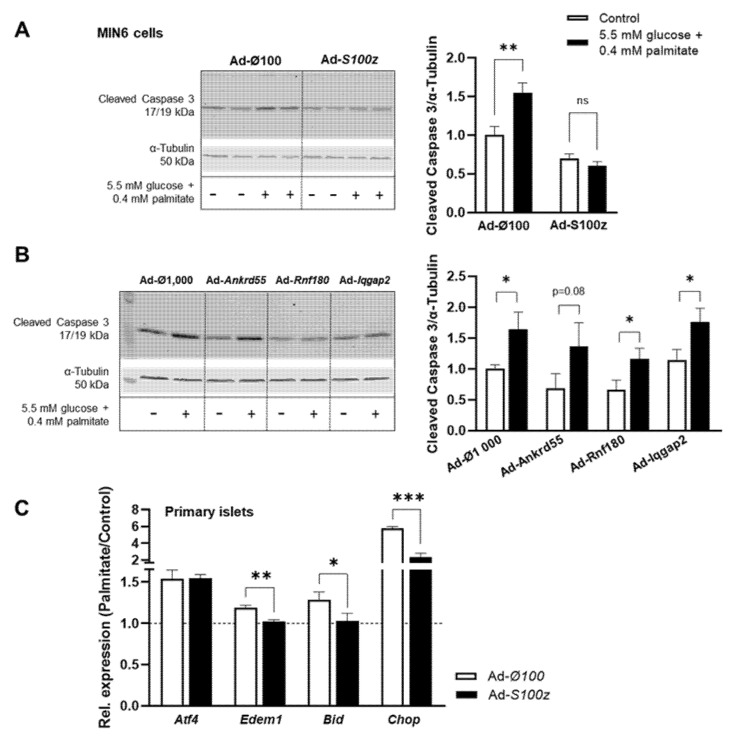
Effect of *Nidd13/NZO* candidate genes on apoptosis. Representative images displaying the effect of MIN6 cells infected with (**A**) Ad-*S100z* vs. Ad-∅100 and (**B**) Ad-*Ankrd55*, Ad-*Rnf180*, and Ad-*Iqgap*2 vs. Ad-∅1000 on protein levels of cleaved caspase-3 as a marker for apoptosis. Infected cells (*n* = 6–8) were stressed with 0.4 mM palmitate for 48 h, and protein levels of cleaved caspase-3 were analyzed and quantified by Western blotting. Here, α-tubulin was used as loading control. (**C**) Primary islet cells of NZO mice were infected with Ad-*S100z* or Ad-∅100 and incubated with 0.4 mM palmitate for 48 h. Expression of ER-stress and apoptotic marker genes were analyzed by qRT-PCR. Data are presented as fold change compared to cells under basal conditions. Data are shown as means ± SEM. * *p* < 0.05; ** *p* < 0.005, *** *p* < 0.001.

## Data Availability

All data used in this manuscript are available upon request.

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
