# Peer review of "Identification of Novel Genes Involved in Hyperglycemia in Mice"

_ijms, 2022, doi:10.3390/ijms23063205_

Round 1
Reviewer 1 Report
The current manuscript is one of multiple follow up studies based on a crossbreeding of the obese and diabetes-prone New Zealand Obese (NZO) mouse strain with four different lean strains (B6, DBA, C3H, 129P2) that vary in their susceptibility to developing T2D to identify novel QTL and genes within which are involved in beta-cell survival and function. Herein the authors characterize and identify the S100z gene, among others, to be implicated in beta-cell proliferation and survival. Although the study is of interest, the authors fail to demonstrate a potential role of this gene in human (and mouse) islet physiology and its potential implication in diabetes.
- As the study is only conducted in mice, the title is misleading. Indeed, mice developed hyperglycemia and not diabetes which is a human disease. As such the title should be changed to reflect that this is a mouse study.
- As the MIN6 cell line is an insulinoma that proliferates and is more resistant to apoptosis, studying such processes in this line may generate flawed results. As such the overexpression studies (Figure 4 and 5) should be repeated in primary mouse and/or human (preferably) islets.
- MIN6 cells are usually cultured in 25 mM glucose. How can 5.6 mM glucose be considered glucotoxic?
Reviewer 2 Report
The manuscript "Identification of novel diabetes-related genes" represents research article, aimed to analyze the effect and the underlying genetic cause of the diabetes QTL (quantitative trait loci) Nidd13/NZO on chromosome 13, a key QTL associated with higher blood glucose levels and increased pancreatic insulin. The authors crossed four mouse inbred strains differing in their diabetes susceptibility with the obese and diabetes-prone NZO strain and identified the QTL Nidd13/NZO on chromosome 13 to correlate with hyperglycemia in NZO allele carriers compared to B6 controls. Subsequent analysis of the critical region, harboring 644 genes, included expression studies in pancreatic islets of con- genic Nidd13/NZO mice, integration of single cell data from parental NZO and B6 islets as well as haplotype analysis. Finally, of five genes (Acot12, S100z, Ankrd55, Rnf180 and Iqgap2) within the polymorphic haplotype block that are differently expressed in islets of B6 compared to NZO mice, they identified the calcium binding protein S100z gene to affect islet cell proliferation as well as apoptosis when overexpressed in MIN6 cells. The authors concluded that they define S100z as most striking gene to be causal for the diabetes QTL Nidd13/NZO by affecting β-cell proliferation and apoptosis as well as an entirely novel diabetes gene regulating islet cell function. Minor issues and suggestions to the authors:
- Please, make this article readable for the readers who are not familiar with all terms which are clear to you only, it will bring wider reading audience to your manuscript
- Please identify the abbreviation QTL (quantitative trait loci) earlier than in line 58
- Please, give explanation of the abbreviation MIN6 cells before line 338
- Line 82, what is HFD
- Please add limitation of the study
Round 2
Reviewer 1 Report
Congrats to the authors for having performed experiments in primary mouse islets which greatly improves the manuscript. I concur that in the age of then pandemic and other world turmoiled, human islets are not easy to procure.